# New composite phenotypes enhance chronic kidney disease classification and genetic associations

Kim Ngan Tran[1], Heidi G. Sutherland[1], Andrew J. Mallett[2,3,4], Lyn R. Griffiths[1], Rodney A. Lea[1]*

1 Centre for Genomics and Personalised Health, Queensland University of Technology, Kelvin Grove, Queensland, Australia, 2 Institute for Molecular Bioscience & Faculty of Medicine, The University of Queensland, St Lucia, Queensland, Australia, 3 Department of Renal Medicine, Townsville University Hospital, Townsville, Queensland, Australia, 4 College of Medicine & Dentistry, James Cook University, Townsville, Queensland, Australia

* rodney.lea@qut.edu.au

## Abstract

Chronic kidney disease (CKD) is a multifactorial condition driven by diverse etiologies that lead to a gradual loss of kidney function. Although genome-wide association studies (GWAS) have identified numerous genetic loci linked to CKD, a large portion of its genetic basis remains unexplained. This knowledge gap may partly arise from the reliance on single biomarkers, such as estimated glomerular filtration rate (eGFR), to assess kidney function. To address this limitation, we developed and applied a novel multi-phenotype approach, combinatorial Principal Component Analysis (cPCA), to better understand the complex genetic architecture of CKD. Using UK Biobank dataset (n = 337,112), we analyzed 21 CKD-related phenotypes, generating over 2 million composite phenotypes (CPs) through cPCA. Nearly 50,000 of these CPs demonstrated significantly higher classification power for clinical CKD compared to individual biomarkers. The top-ranked CP—a combination of albumin, cystatin C, eGFR, gamma-glutamyltransferase, HbA1c, low-density lipoprotein, and microalbuminuria, achieved an AUC of 0.878 (95% CI: 0.873–0.882), significantly outperforming eGFR alone (AUC: 0.830, 95% CI: 0.825–0.835). Genetic association analysis of the ~ 50,000 high-performing CPs identified all major eGFR-associated loci, except for the *SH2B3* locus rs3184504, a loss-of-function variant, which was uniquely identified in CPs (p = 3.1 × 10^{-56}) but not in eGFR within the same sample size. In addition, *SH2B3* locus showed strong evidence of colocalization with eGFR, supporting its role in kidney function. These results highlight the power of the multi-phenotype cPCA approach in understanding the genetic basis of CKD, with potential applications to other complex diseases.

**Data availability statement:** The summary statistics are publicly available in Figshare repository at https://doi.org/10.6084/m9.figshare.26122540.v1. All the scripts used in the study are available in the S2 Text.

**Funding:** This work was supported by Queensland University of Technology (to KNT) and the Queensland Health (to AJM). The funders had no role in study design, data collection and analysis, decision to publish, or preparation of the manuscript.

**Competing interests:** The authors have declared that no competing interests exist.

## Author summary

Chronic kidney disease (CKD) can result from diverse underlying causes, such as diabetes, high blood pressure, infections, and lifestyle factors. However, most CKD studies rely on single measurements, such as estimated glomerular filtration rate (eGFR), which assesses kidney filtration but may not fully capture the complexity of the disease. Here, we applied a novel approach to explore CKD from a broader perspective. Using a large dataset of over 300,000 individuals, we combined 21 kidney-related health measures into millions of new composite traits, providing a more comprehensive view of kidney function. One of these composite traits resulted from a combination of albumin, cystatin C, eGFR, gamma-glutamyltransferase, HbA1c, low-density lipoprotein, and microalbuminuria, proved to be significantly more effective at identifying CKD than any single measurement. Additionally, we identified key genetic factors associated with CKD, including the *SH2B3* gene. By integrating multiple measurements, our work offers a clearer understanding of the genetic basis of CKD and paves the way for similar approaches to unravel other complex diseases, ultimately aiding in their prevention and treatment.

## Introduction

Chronic kidney disease (CKD) is a collective term encompassing a range of heterogeneous diseases characterized by persistent structural or functional kidney abnormalities. CKD is stratified into five stages, culminating in kidney failure, which necessitates consideration of interventions such as kidney transplantation or dialysis. This condition has a high prevalence, affecting approximately 10–15% of the global population, resulting in significant burden on both public health and the economy [1].

Genome-wide association studies (GWAS) investigating CKD have traditionally focused on evaluating kidney function using single biomarkers, such as estimated glomerular filtration rate (eGFR), microalbuminuria, or blood urea nitrogen [2–5]. For example, a robust GWAS analysis of eGFR in a cohort of over 1.2 million individuals identified 634 independent genetic signals, collectively accounting for 9.8% of the eGFR variance [4]. However, a portion of the heritability of CKD remains unexplained. This gap in understanding can be attributed, in part, to the fact that eGFR and other individual biomarkers do not fully capture the underlying causes of CKD nor accurately predict an individual's risk of CKD or progression to kidney failure [6]. For a comprehensive diagnosis and prognosis of systemic CKD, it is recommended to employ a combination of various markers that collectively reflect the diverse alterations occurring over the course of CKD development [7].

Previously, we employed principal component analysis (PCA) on multiple quantitative phenotypes associated with CKD, uncovering a novel susceptibility gene for kidney function that remained undetected in single-phenotype GWASs [8]. In this study, we introduce and implement a new multi-phenotype approach termed combinatorial PCA to further investigate the genetic basis of CKD within the UK Biobank dataset.

## Results

In this study, our objective was to identify novel genetic loci associated with CKD through a comprehensive multi-phenotype analysis. We conducted our analyses on the White-British group within the UK Biobank (UKB) prospective cohort study (n = 337,112) to identify composite phenotypes (CPs) with higher performance in CKD classification compared to individual CKD-related biomarkers. Subsequently, we performed GWAS on the identified CPs and replicated the results in the Irish cohort, also within the UKB (n = 11,106).

### Best single-markers for CKD classification

Prior to conducting the multi-phenotype analysis, we examined the 21 phenotypes previously linked to CKD (Table 1) in terms of their performance in classifying clinical CKD. This was evaluated by the area under the curve (AUC) of receiver operating characteristic (ROC) curves using the ICD codes for CKD as clinical outcomes. The biomarkers encompassed a range of physiological indicators of CKD risk, including markers of renal function, metabolic parameters, inflammation, lipid profile, and blood pressure. eGFR and CYSC both had the highest discriminatory power among the biomarkers, exhibited by the highest AUCs ranging from 0.825 to 0.842. Although the AUC value of CYSC was slighly higher than that of eGFR, comparing the 2 ROC curves showed no statistically significant difference (p = 0.256). Other biomarkers, such as blood urea nitrogen (BUN), uric acid (UA), and glycated hemoglobin (HbA1c), demonstrated moderate discriminatory performance, with AUCs ranging from 0.658 to 0.742. Conversely, other biomarkers such as vitamin D (VITD), calcium (CALC), and diastolic blood pressure (DBP) exhibited low AUC values, ranging from 0.489 to 0.525.

### Composite phenotypes perform better than single markers in CKD classification

In this multi-phenotype analysis, we developed and applied a method called combinatorial PCA (cPCA) to identify combinations of biomarkers that outperformed single markers. General steps of the cPCA method are illustrated in Fig 1A. Through cPCA application, a total of 2,097,130 composite phenotypes (CPs) were extracted from all unique combinations of 21 CKD-related biomarkers, and were subsequently evaluated for performance in CKD classification. As a result, we identified 49,734 CPs with significantly better disease classification compared to eGFR as a single biomarker ($p < 2.5e \times 10^{-8}$, as adjusted for 2 million independent tests), as assessed using ROC curves and AUC. The top ten CPs with the highest performance are listed in Table 2.

We analyzed the phenotypic components of the 49,734 CPs that exhibited statistically significantly better performance in CKD classification compared to eGFR (Fig 1B, 1C and 1D). The top ranked CP was represented by albumin (ALB), CYSC, eGFR, gamma glutamyl-transferase (GGT), HbA1c, low density lipoprotein (LDL), and microalbuminuria (MA) (AUC = 0.878, 95%CI = 0.873-0.882). Among the other combinations, CYSC and eGFR were consistently present, with HbA1c appearing in nearly all instances. Other notable phenotypes included MA, ALB, and LDL, with appearances ranging from 75% to 55% across the combinations. Regarding pairs or triples of phenotypes, as expected, the most frequent combinations included CYSC, eGFR, and HbA1c: CYSC-eGFR pairs were present in all combinations, while CYSC-HbA1c, eGFR-HbA1c, and CYSC-eGFR-HbA1c were found in 99% of combinations.

We analyzed the relationship between the CP extracted from {eGFR, CYSC, ALB, HbA1c, GGT, LDL, and MA} and CKD using a logistic regression model: CKD status ~ CP + age + sex. In this model, the odds ratio (OR) for CP was 0.419 (95% CI: 0.413 – 0.425), indicating that each one-unit increase in CP corresponds to a 58.1% decrease in CKD odds. Fig 2 presents the PCA biplot, while the loadings of each biomarker contributing to the CP are detailed in S1 Table.

### Genetic associations of the identified CPs

The cPCA analysis identified a total of 49,734 CPs with significantly higher AUCs than that of eGFR. Although each of the CPs might have distinct underlying genetic bases due to their constituent biomarkers, they all shared a common

**Table 1. Twenty one kidney function related phenotypes selected from the UK Biobank dataset.**

| No. | Phenotypes | Abbr. | Relation to kidney function | AUC | 95% CI |
|---|---|---|---|---|---|
| 1 | Cystatin C | CYSC | CYSC levels associated with kidney function [9]. | 0.837 | 0.832-0.842 |
| 2 | Creatinine-based eGFR (CKD-EPI Creatinine Equation - 2021) | eGFR | Marker of kidney function [10]. | 0.830 | 0.825-0.835 |
| 3 | Urea | UREA | Higher UREA levels associated with adverse renal outcomes [11]. | 0.735 | 0.729-0.742 |
| 4 | Urate | UA | Higher UA associated with new and progressive CKD [12]. | 0.681 | 0.675-0.687 |
| 5 | Glycated haemoglobin | HbA1c | Higher HbA1c associated with increased risk of CKD or CVD [13]. | 0.664 | 0.658-0.67 |
| 6 | C-reactive protein | CRP | Higher CRP associated with CKD incidence [14]. | 0.634 | 0.628-0.64 |
| 7 | Body mass index | BMI | Higher BMI associated with increased risk of CKD [15–17]. | 0.631 | 0.625-0.637 |
| 8 | LDL direct | LDL | Higher LDL associated with increased risk of CVD in non-dialysis CKD patients [18]. | 0.628 | 0.621-0.635 |
| 9 | HDL cholesterol | HDL | Both low and high HDL associated with adverse outcomes in patients with CKD [19]. | 0.628 | 0.621-0.634 |
| 10 | Apolipoprotein B | APOB | Higher APOB associated with lower eGFR, increased ESRD risk [20–22]. | 0.602 | 0.595-0.609 |
| 11 | Apolipoprotein A | APOA1 | Higher APOA1 associated with lower CKD prevalence [23]. | 0.600 | 0.593-0.606 |
| 12 | Albumin | ALB | Associated with reduced kidney functions in HIV-infected individuals and elders [24,25]. | 0.590 | 0.584-0.597 |
| 13 | Triglycerides | TRIG | Associated with CKD stages [26]. | 0.588 | 0.582-0.594 |
| 14 | Gamma glutamyltransferase | GGT | Higher GGT associated with increased risk of ESRD [27]. | 0.582 | 0.576-0.589 |
| 15 | Systolic blood pressure | SBP | Lower SBP associated with ESRD and increased mortality in CKD patients [28]. | 0.572 | 0.566-0.579 |
| 16 | Microalbuminuria | MA | Biomarker for kidney injury [29]. | 0.564 | 0.56-0.568 |
| 17 | Haematocrit percentage | HCT | Lower HCT associated with declined kidney function and increased risk of ESRD [30,31]. | 0.549 | 0.542-0.556 |
| 18 | Phosphate | PHOS | High PHOS associated with increased CVD risk and mortality in patients with or without CKD [32]. | 0.521 | 0.515-0.528 |
| 19 | Vitamin D | VITD | Lower VITD associated with adverse outcomes and mortality in CKD patients [33]. | 0.518 | 0.512-0.525 |
| 20 | Calcium | CALC | Lower CALC associated with rapid CKD progression [34]. | 0.505 | 0.498-0.512 |
| 21 | Diastolic blood pressure | DBP | Lower DBP associated with increased mortality in CKD patients [28,35]. | 0.496 | 0.489-0.503 |

ESRD: end-stage renal disease.

kidney-function-related genetic component, owing to their superior discriminatory power in CKD classification compared to eGFR. To identify the shared genetic loci associated with kidney function across these CPs, we looked for loci that consistently reached genome-wide significance (p < 5×10⁻⁸) across all 49,734 CPs.

This analysis yielded 80 loci consistently identified in all the identified CPs (p = 5×10⁻⁸) and as shown in Fig 3. most of these loci were also observed in our eGFR GWAS. However, 5 loci – *CST3, SH2B3*, *FTO*, *SEMA3F-AS1*, and *AC128707.1* - were not identified in the eGFR GWAS and were instead found in GWASs of other individual phenotypes. *SH2B3* was found in 12 out of the 21 individual-phenotype GWASs, *FTO* was found in 7 and *SEMA3F-AS*1 in 5, and *AC128707.1* was found to be genome-wide significant in 2 GWASs, and *CST3* was only found to be genome-wide significant in the GWAS of CYSC.

Among these loci, 75 were found to overlap with those identified in the GWAS of eGFR, while 5 loci were not. Instead, these 5 loci were discovered in GWASs of other individual phenotypes, each represented by a distinct color.

 

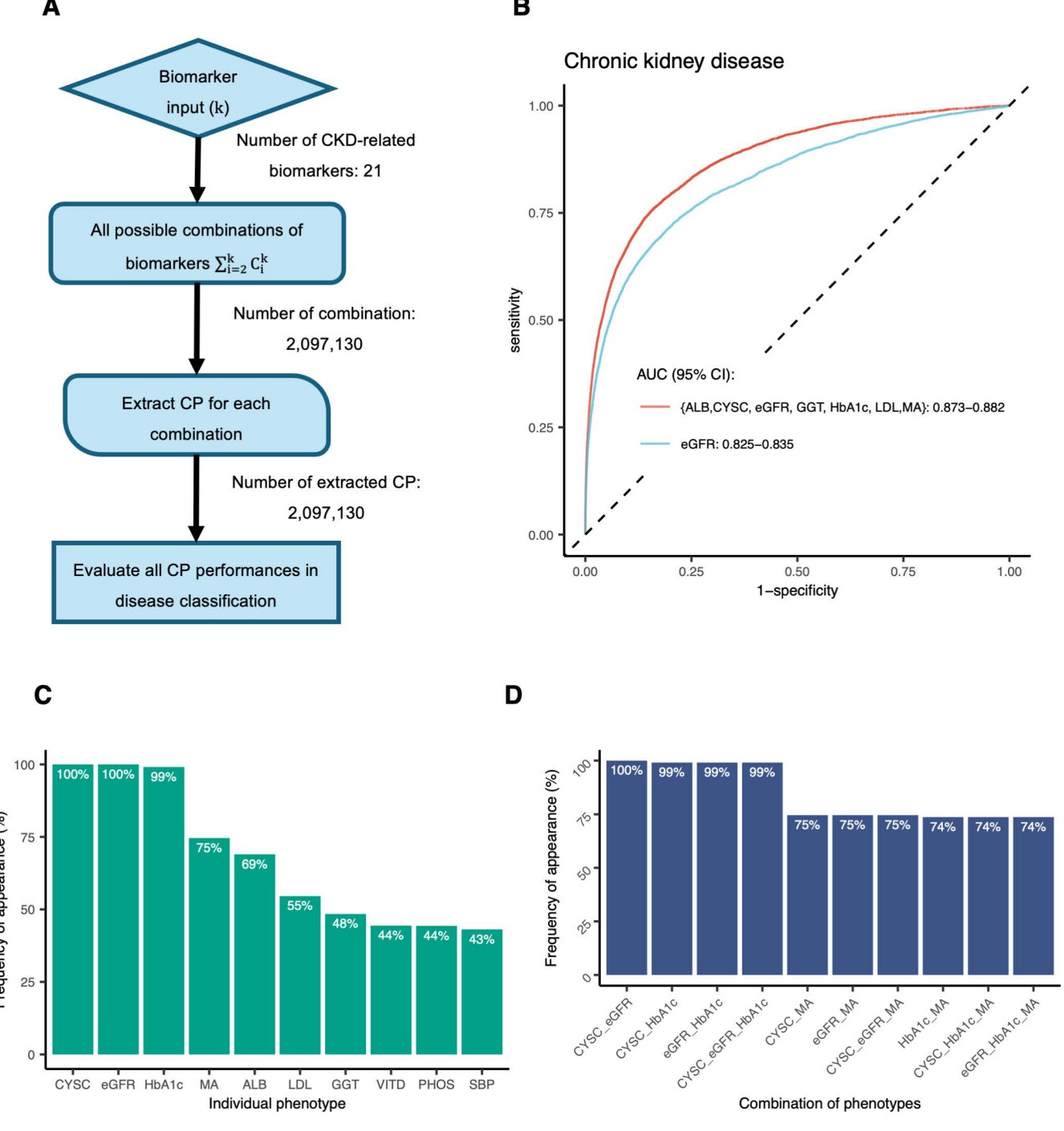

**Fig 1. Combinatorial Principal Component Analysis (cPCA).** (A) Flowchart of the cPCA method. (B) The ROC curve of the top CP extracted from eGFR, CYSC, MA, HbA1c, LDL, ALB, and GGT in comparison to the ROC curve of eGFR in terms of CKD classification. (C) Top 10 of the phenotypes that appeared most frequently in the ~50,000 CPs. (D) Top 10 of the phenotypes pairs or triples that appeared most frequently in the ~50,000 CPs.

**Table 2. Top 10 CPs with the highest AUCs for CKD classification.**

| No. | Combination which CP extracted from | AUC | 95% CI | P-values |
|---|---|---|---|---|
| 1 | {ALB, CYSC, eGFR, GGT, HbA1c, LDL, MA} | 0.878 | 0.873-0.882 | $3.7 \times 10^{-149}$ |
| 2 | {ALB, CYSC, eGFR, GGT, HbA1c, LDL, MA, PHOS} | 0.878 | 0.873-0.882 | $3.4 \times 10^{-146}$ |
| 3 | {ALB, CYSC, eGFR, GGT, HbA1c, LDL, MA, PHOS, VITD} | 0.877 | 0.873-0.882 | $9.1 \times 10^{-141}$ |
| 4 | {ALB, CYSC, eGFR, GGT, HbA1c, LDL, MA, VITD} | 0.877 | 0.873-0.882 | $2.6 \times 10^{-140}$ |
| 5 | {ALB, CALC, CYSC, DBP, eGFR, GGT, HbA1c, LDL, MA} | 0.876 | 0.872-0.881 | $1.2 \times 10^{-134}$ |
| 6 | {ALB, CYSC, eGFR, HbA1c, LDL, MA, PHOS, SBP} | 0.876 | 0.872-0.881 | $8.4 \times 10^{-142}$ |
| 7 | {ALB, CALC, CYSC, eGFR, HbA1c, LDL, MA, PHOS, SBP} | 0.876 | 0.872-0.881 | $5.5 \times 10^{-137}$ |
| 8 | {ALB, CALC, CYSC, eGFR, HbA1c, LDL, MA, SBP} | 0.876 | 0.872-0.88 | $2.9 \times 10^{-138}$ |
| 9 | {ALB, APOB, CYSC, eGFR, GGT, HbA1c, MA} | 0.876 | 0.872-0.88 | $2.6 \times 10^{-145}$ |
| 10 | {ALB, CALC, CYSC, DBP, eGFR, GGT, HbA1c, LDL, MA, VITD} | 0.876 | 0.872-0.88 | $1.3 \times 10^{-135}$ |

Each CP's ROC curve was compared with the eGFR's ROC curve (AUC = 0.830, 95% CI: 0.825-0.835) and P-value was derived based on the boostrap method with 2000 resamples.

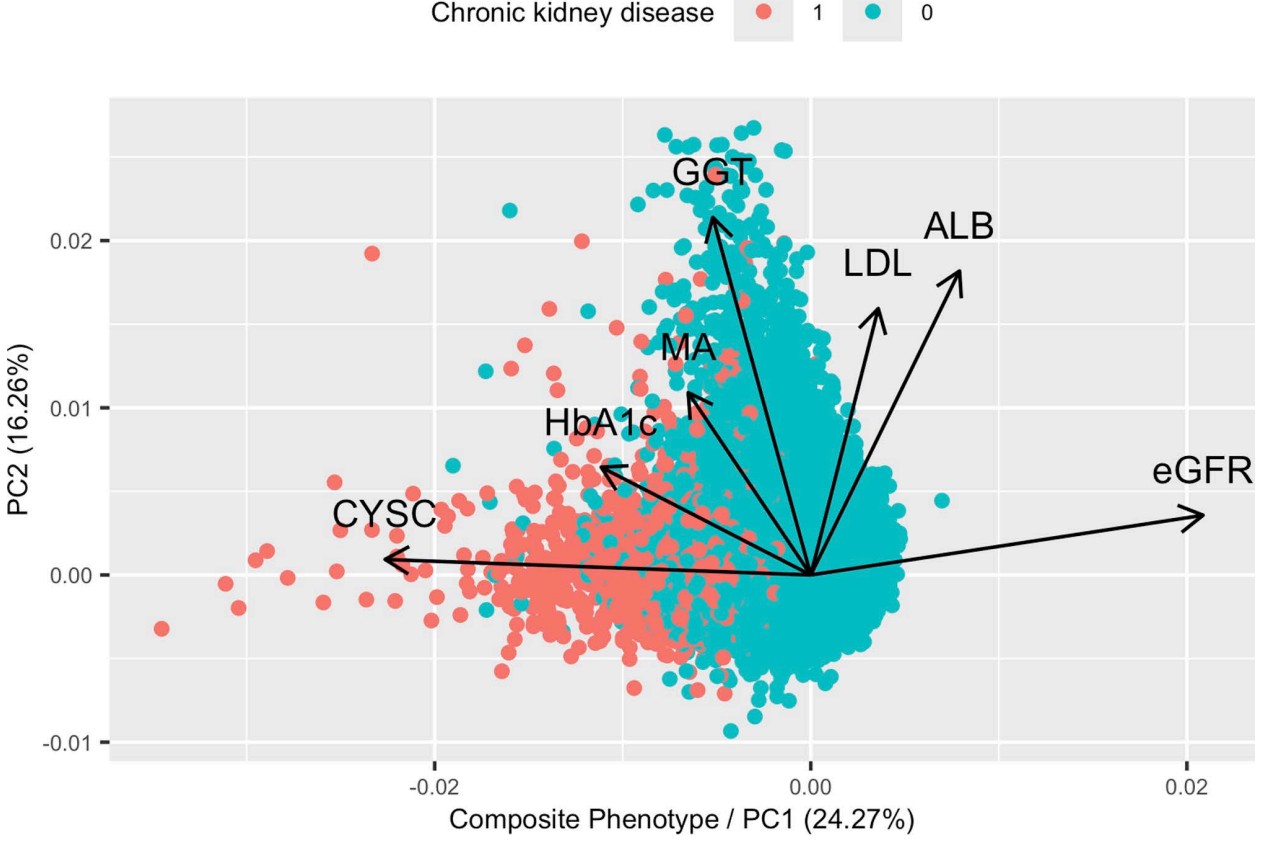

**Fig 2. PCA illustrating the PCA scores and the trait loadings on the CP extracted from eGFR, CYSC, MA, HbA1c, LDL, ALB, and GGT.** Samples were labeled with chronic kidney disease status (1:case; 0:non-case).

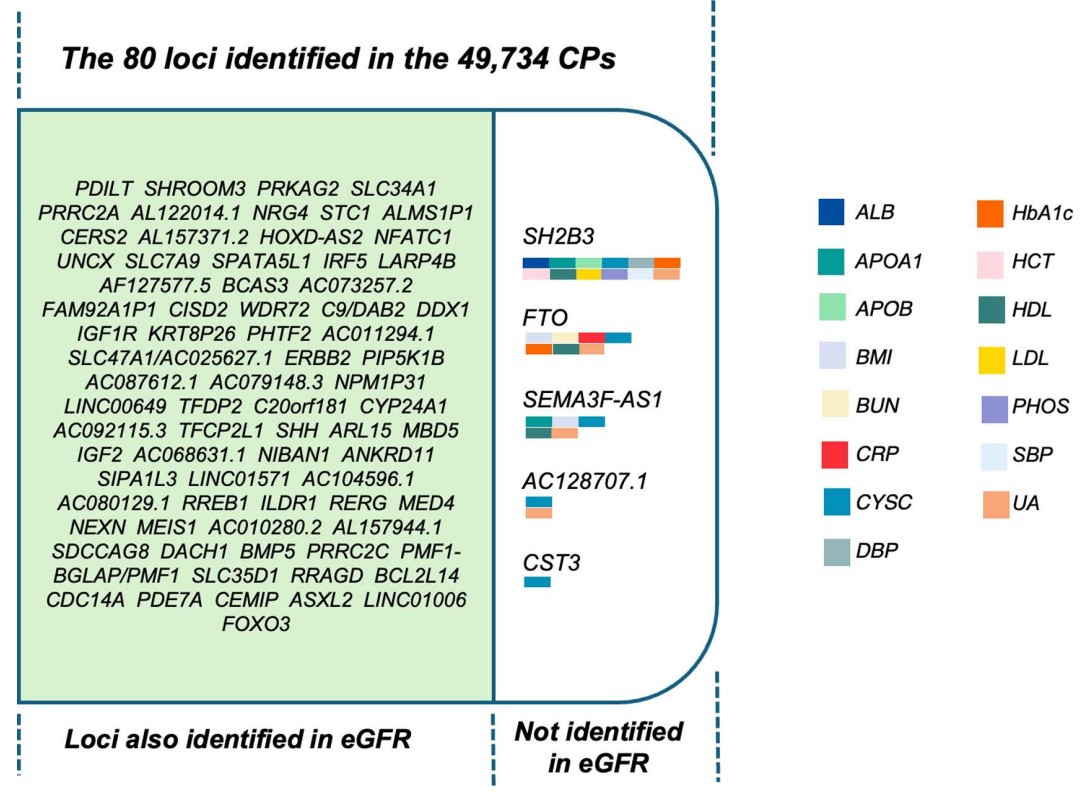

**Fig 3. The 80 genetic loci identified in all the ~50,000 CPs.**

## Replication in Irish cohort and validation against large-scale eGFR GWAS

We utilised the independent cohort of Irish ethnicity in the UKB dataset for our replication analysis. In the discovery group, i.e., the British cohort, the CP extracted from the combination of {eGFR, CYSC, ALB, HbA1c, GGT, LDL, and MA} was among those that had the highest AUCs for CKD classification and at the same time had the least number of phenotypes (Table 2). Therefore, we selected this combination of phenotypes to generate a new CP for the replication cohort. As a result, the new CP in the replication cohort also had significantly better performance in CKD classification compared to those of individual phenotypes (Fig 4). Out of the 5 loci identified through the multi-phenotype approach, *CST3* and *SH2B3* were replicated in the Irish cohort as outlined in Table 3.

The CP was extracted from the combinations of phenotypes {eGFR, CYSC, ALB, HbA1c, GGT, LDL, and MA}.

In addition to the replication in the independent Irish cohort, we also looked at the genetic associations with eGFR in larger GWAS to examine whether these identified loci could be identified. We compared them against the large-scale GWAS of creatinine-based eGFR conducted in 1.2 million individuals [4]. Overall, the effect directions were consistent across the five loci (Table 4). However, only the *SH2B3* locus reached genome-wide significance ($p \leq 5 \times 10^{-8}$). Although not genome-wide significant, the *SEMA3F-AS1* and *AC128707.1* loci reached suggestive significance thresholds ($p \leq 1 \times 10^{-5}$), while the *CST3* and *FTO* loci showed no evidence of association ($p \geq 0.05$).

## Colocalization analysis

To further investigate the genetic architecture underlying the composite phenotypes, we conducted a colocalization analysis to determine whether the loci identified through the multi-phenotype cPCA approach share causal variants with known

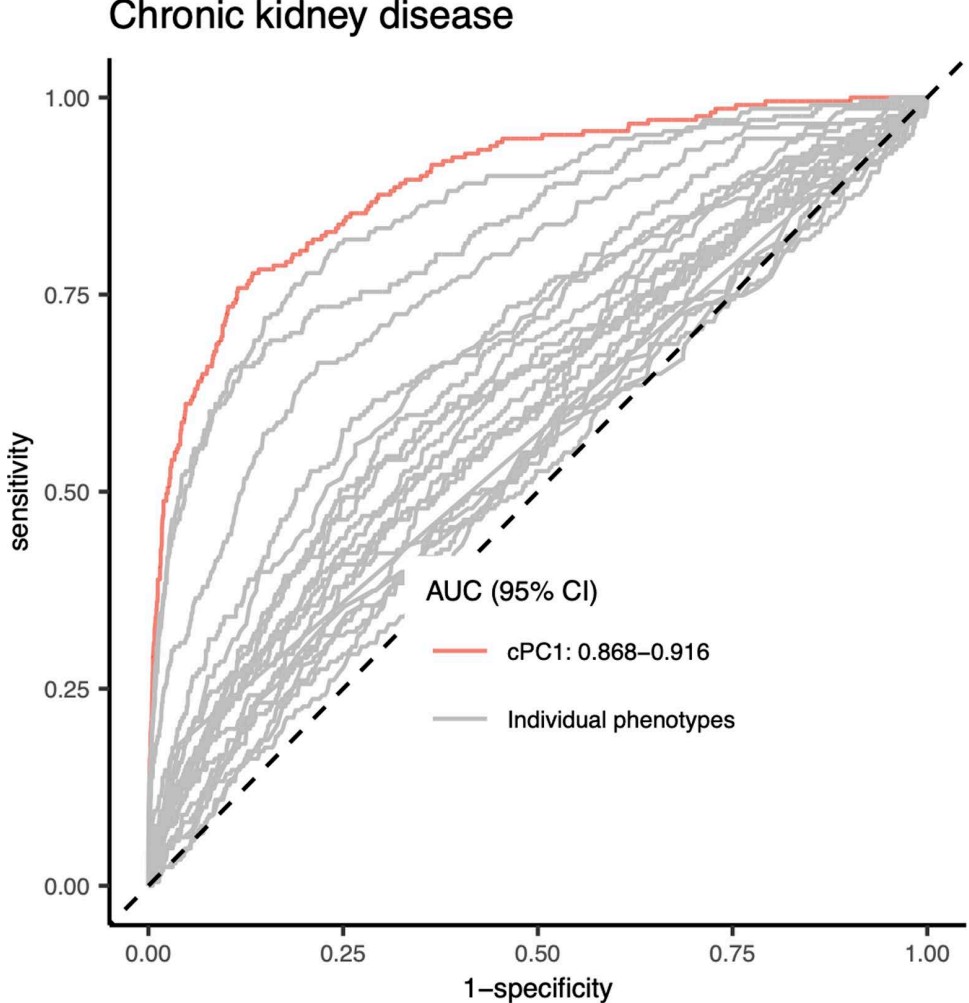

**Fig 4. ROC curves for CKD classification of the CP and the 21 CKD-related phenotypes in the replication Irish cohort.**

**Table 3. Association results of the identified kidney-function loci in the discovery British group and the replication Irish group.**

| No. | Gene | Chr. | Position | rsID | A1 | A2 | British (n = 296,372) | | | Irish (n = 11,206) | | |
|---|---|---|---|---|---|---|---|---|---|---|---|---|
| | | | | | | | Beta | SE | P | Beta | SE | P |
| 1 | CST3 | 20 | 23,569,186 | rs2405392 | T | C | 0.190 | 0.003 | 0 | 0.223 | 0.018 | $1.90 \times 10^{-35}$* |
| 2 | SH2B3 | 12 | 111,884,608 | rs3184504 | T | C | -0.045 | 0.003 | $7.15 \times 10^{-53}$ | -0.061 | 0.015 | $5.74 \times 10^{-5}$* |
| 3 | FTO | 16 | 53,818,834 | rs56313538 | G | A | -0.026 | 0.003 | $2.30 \times 10^{-18}$ | -0.037 | 0.016 | 0.017 |
| 4 | SEMA3F-AS1 | 3 | 50,174,197 | rs2624847 | G | T | -0.024 | 0.003 | $6.76 \times 10^{-13}$ | 0.011 | 0.017 | 0.505 |
| 5 | AC128707.1 | 12 | 78,807,411 | rs7311712 | T | C | 0.019 | 0.003 | $1.22 \times 10^{-10}$ | -0.004 | 0.015 | 0.800 |

The phenotypic outcomes were CP extracted from {eGFR, CYSC, ALB, HbA1c, GGT, LDL, and MA} for both the British group and the Irish group, respectively.

* p-values < 0.01 as accounted for multiple corrections.

**Table 4. Validation against the large GWAS of creatinine-based eGFR (n = 1,201,909).**

| No. | Gene | Chr. | Position | rsID | A1 | A2 | GWAS meta-analysis for eGFR | | |
|---|---|---|---|---|---|---|---|---|---|
| | | | | | | | Beta | SE | P |
| 1 | CST3 | 20 | 23,569,186 | rs2405392 | T | C | $6.00 \times 10^{-4}$ | $3.00 \times 10^{-4}$ | 0.04875 |
| 2 | SH2B3 | 12 | 111,884,608 | rs3184504 | T | C | -0.0016 | $3.00 \times 10^{-4}$ | $6.72 \times 10^{-10}$ |
| 3 | FTO | 16 | 53,818,834 | rs56313538 | G | A | $-2.00 \times 10^{-4}$ | $3.00 \times 10^{-4}$ | 0.534 |
| 4 | SEMA3F-AS1 | 3 | 50,174,197 | rs2624847 | G | T | -0.0014 | $3.00 \times 10^{-4}$ | $4.86 \times 10^{-6}$ |
| 5 | AC128707.1 | 12 | 78,807,411 | rs7311712 | T | C | 0.0013 | $3.00 \times 10^{-4}$ | $5.70 \times 10^{-7}$ |

eGFR-associated loci. This analysis aimed to validate the relevance of the identified loci to kidney function and assess their potential pleiotropic effects.

Among the two loci uniquely identified using cPCA and successfully replicated—CST3 and SH2B3—only SH2B3 showed strong evidence of colocalization with eGFR GWAS signals (Table 5). In contrast, CST3 was associated with composite phenotypes but did not share a causal variant with eGFR, suggesting its role as a biomarker rather than a direct causal gene for CKD.

## Discussion

CKD is a common term to describe a range of diseases characterized by impaired kidney structure, or reduced kidney function over time. Because there is an incomplete understanding of the genetics for different CKD subtypes, the identification of effective drug targets has been hindered. Research has tended to focus on eGFR or other single CKD-related biomarkers, yet this approach could be inadequate for capturing the underlying CKD etiology or pathophysiology. Since CKD is associated with many individual phenotypes, we reasoned that a multi-phenotype analytical approach may identify novel genetic loci relevant to CKD. Specifically, we designed a combinatorial PCA algorithm (cPCA), the aim of which was to extract relevant composite phenotypes for accurate CKD classification. This involved iteratively exploring all possible combinations of the 21 input phenotypes to identify composite phenotypes that outperformed individual biomarkers in CKD classification. Over 2 million phenotypic combinations were analyzed, resulting in the identification of nearly 50,000 composite phenotypes with significantly higher AUCs than eGFR or any individual phenotype.

The primary objectives of cPCA are to identify optimal combinations of biomarkers that collectively enhance disease classification, outperforming individual biomarkers alone, and to identify genetic loci associated with the target disease by leveraging multi-trait GWAS approaches. Regarding the second objective, cPCA shares similarities with multivariate GWAS and other joint analyses of multiple traits, including PCA-based methods (e.g., combined-PC [36] or adaptive principle component test [37]). However, unlike traditional multivariate methods that require a predefined set of traits, cPCA systematically explores and selects optimal biomarker combinations, making it a distinct and more flexible approach.

CYSC, eGFR, HbA1c, MA, ALB, LDL, and GGT were the most frequently observed phenotypes, appearing in 75% to 48% of those combinations. The frequent presence of HbA1c, ALB, LDL, and GGT alongside well-established CKD

**Table 5. Colocalization probabilities for each locus are presented across five categories.**

| Loci | No causal variant | Causal variant for the CP only | Causal variant for eGFR only | Two distinct causal variants | One common causal variant |
|---|---|---|---|---|---|
| SH2B3 | $1.39 \times 10^{-39}$ | 0.1745 | $7.12 \times 10^{-41}$ | 0.008 | **0.817** |
| CST3 | $3.41 \times 10^{-291}$ | **0.970** | $3.66 \times 10^{-293}$ | 0.010 | 0.020 |

Probabilities range from 0 to 1, with higher values indicating stronger support for a given colocalization scenario.

phenotypes such as CYSC, eGFR, and MA highlighted the overlap between kidney function and other aspects of human health, including blood glucose levels, cardiovascular health, and liver function [38–40]. Furthermore, we observed that although BUN and UA, which are highly correlated with eGFR, exhibited higher performance in CKD classification than HbA1c, MA, and others, they appeared less frequently in the ~50,000 combinations (BUN: 22.4% and UA: 0%). This suggests that cPCA could mitigate multicollinearity by ensuring the inclusion of independent phenotypes that are not highly correlated.

Consequently, we analysed the genetic associations and identified 80 loci that were consistently reached genome-wide significant in all the ~50,000 CPs. There were 5 loci that were not identified in eGFR but consistently identified in all the ~50,000 CPs. They were *CST3, SH2B3*, *FTO*, *SEMA3F-AS1*, and *AC128707.1*. *CST3* and *SH2B3* were successfully replicated in an independent cohort. *CST3*, encoding Cystatin-C (CYSC), a key marker of kidney function, appears to have been driven primarily by the presence of CYSC in all the combinations (Fig 1C). As *CST3* was not found in any other individual-trait GWAS except for CYSC's, it is more likely to be a biomarker gene than a causal gene influencing CKD pathology.

By contrast, *SH2B3*, encoding a cytokine-signaling regulator, was found to be genome-wide significant in a total of 12 single-trait GWASs including those for ALB, APOA1, APOB, CYSC, DBP, HbA1c, HCT, HDL, LDL PHOS, SBP, and UA. This was consistent with the fact that the index SNP rs3184504, which is also a missense SNP of *SH2B3*, has been found to be associated with multiple phenotypes and diseases relating to blood pressure, blood cells, cholesterol levels, as well as cardiovascular diseases and type-1 diabetes. The effect allele T had an effect size of -0.045 (SE = 0.003) in the GWAS of CP extracted from {eGFR, CYSC, ALB, HbA1c, GGT, LDL, and MA} (Tables 3 and 4), indicating that each additional copy of the T allele is associated with a decrease in CP. Since higher CP values correspond to lower CKD odds, this suggests that the *SH2B3* variant increases CKD risk by reducing CP. This finding is consistent with previous studies showing that animal models homozygous for the T allele, generated using CRISPR-Cas9, exhibited higher blood pressure and exacerbated kidney dysfunction compared to control mice [41].

Although it was not genome-wide significant in our eGFR GWAS (beta = -0.00856, p = 8.47$\times$10-5, n = 337,112), rs3184504-T has been shown to reach significance in larger GWAS cohorts (beta = -0.0016, p = 5.3$\times$10-8, n = 1,031,620) [4]. In our study this variant was robustly identified using a comparatively smaller cohort (~300,000 subjects) and successfully replicated in an even smaller sample size (n = 11,206). This finding emphasizes the increased power and effectiveness of the method (cPCA) in identifying genetic associations for complex phenotypes like CKD.

However, the cPCA approach also has its limitations. While it could identify genes with strong discriminatory power, it struggled to differentiate between biomarkers and causal genes. *CST3* is likely a biomarker reflecting kidney function rather than a gene driving CKD pathology, whereas *SH2B3* appears to play a more direct functional role. This limitation stemmed from the inclusion of biomarkers that reflect disease status but may not be directly involved in the underlying disease mechanisms (e.g., Cystatin-C).

Another limitation of this study is the reliance on ICD codes for identifying CKD outcomes. ICD codes often exhibit low sensitivity in detecting CKD, leading to potential underreporting of cases [42]. For instance, a study found that ICD-10 codes had a sensitivity ranging from 25% to 51% for detecting CKD stages G3-5, depending on the specific codes used [43]. This underreporting may introduce selection bias toward individuals with more severe disease, thereby limiting the generalizability of our findings. Nevertheless, the use of ICD-based CKD diagnoses remains common in large-scale biobank studies, particularly when serum creatinine-based eGFR values are unavailable for long-term CKD classification. While this approach likely enriches the cohort for more advanced CKD cases, it maintains high specificity—reported between 82% and 99%—ensuring that cases included in our analysis truly represent CKD rather than transient reductions in kidney function [43,44].

In conclusion, our multi-phenotype approach highlighted the increased power of cPCA in identifying CKD-related loci. Future studies should incorporate functional validation and explore a broader range of phenotypes to better understand the genetic architecture of CKD.

## Methods

### Ethics statement

Ethical approval for the UKB study was obtained from the North West Multi-Centre Research Ethics Committee, and all participants provided written informed consent. This research has been conducted utilizing the UK Biobank Resource under Application Number 86460.

### Research cohort

The UK Biobank (UKB) [45] is a longitudinal cohort study designed to investigate the interplay between genes, the environment, and health. The study includes over 500,000 participants aged 40–69 years, recruited between 2006 and 2010 from 22 assessment centers across England, Scotland, and Wales. Participants provided detailed information about their health and lifestyle, donated samples of blood, urine, and saliva for long-term storage and analysis, and underwent various physical measurements, including height, weight, spirometry, blood pressure, and heel bone density.

We selected White-British samples, constituting the largest ethnic group within the UKB dataset, for the discovery cohort based on both the 'Ethnic background' and 'Genetic ethnic grouping' data. This approach allowed us to accurately identify individuals who self-identified as 'White British' and exhibited very similar genetic ancestry profiles, as determined by a principal components analysis of their genotypic data. Additionally, we excluded individuals whose genetic sex differed from their self-identified sex, those with sex chromosome aneuploidy, or those who were not included in the genetic principal components analysis conducted by the UKB research team. The final sample size was 337,112. Finally, we included individuals from the Irish ethnicity within the UKB dataset for the replication analyses (n = 11,106). The data processing steps were performed similarly to those used for the discovery cohort.

### Phenotype data

In total, 21 biomarkers relevant to chronic kidney disease (CKD) were included in this study (Table 1). These phenotypes were assessed based on the correlations of the measurements with prevalence of CKD, CKD stages, kidney function, and an increased risk of adverse outcomes in individuals with CKD. All measurements were collected at baseline for all participants. Details of the assay manufacturers, analytical platforms, and analysis methodologies can be found at https://www.ukbiobank.ac.uk/enable-your-research/about-our-data/biomarker-data. Quantitative measures outside their respective analytical ranges were treated as missing data. For urine albumin, which is essential for calculating the urine albumin-to-creatinine ratio (UACR), we implemented a multi-step procedure to address cases where urine albumin levels fell below the detection threshold of 6.7 mg/L for a significant number of participants (see S1 Text). Estimated GFR (eGFR) was calculated using the creatinine-based CKD-EPI-2021 equation without race coefficient. [10] Samples with more than 30% missing data points were excluded. Remaining missing phenotypic values were imputed to obtain a complete dataset using the R package missMDA v1.11 [46], ensuring that the imputed values had no effect on the principal component analysis (PCA) results. PCA was performed using the R package FactoMineR v1.34 [47].

### Genotype data

Genome-wide genotyping was conducted on all UKB participants using the UK Biobank Axiom Array. Approximately 850,000 variants were directly measured, while over 90 million variants were imputed using the Haplotype Reference Consortium and UK10K + 1000 Genomes reference panels. Imputation data were stored in the compressed and indexed BGENv1.2 format. We converted the data from BGEN format into binary PGEN files and performed quality control procedures within PLINK2.0 [48]. The criteria for selecting variants were: (1) autosomal variants; (2) missing rate of less than 5%; (3) not significantly deviated from Hardy-Weinberg equilibrium (p-value = $10 \times 10\text{-}10$); (4) minor allele frequency (MAF) of at least 0.01; and (5) imputation score of more than 0.8. After quality control, we retained 12.7 million SNPs for subsequent analysis.

PLOS Genetics

## CKD clinical outcome data

Health-related outcome data are available in death, hospital, and primary care records. Using the ICD-10 and ICD-9 codes (International Classification of Diseases, tenth and ninth editions), we categorized individuals diagnosed with chronic kidney disease, renal failure, renal sclerosis, chronic glomerulonephritis, nephritis, nephropathy, hypertensive chronic kidney disease, hypertensive heart and kidney disease, diabetes with renal complications, kidney replaced by transplant, disorders resulting from impaired renal function, or unspecified disorders of the kidney and ureter as CKD cases.

## Combinatorial principal component analysis (cPCA)

**Principles:** We developed an multi-phenotype approach called combinatorial PCA (cPCA) to identify combinations of bio-markers that collectively offer improved discriminatory power in disease classification compared to individual biomarkers alone. In cPCA, various combinations with varying numbers of biomarkers are generated from a fixed set of input biomark-ers. The number of possible combinations generated can be calculated as $\sum_{i=2}^{k} C_i^k$. The first principal component, denoted as CP, is then extracted to represent each combination. CP serves as a comprehensive biomarker signature, representing the maximum variance direction within the biomarker combination. Finally, the performance of each CP in disease classifi-cation is evaluated and compared to that of single biomarkers.

**Implementation Details:** To systematically explore and identify potential superior components for CKD classifica-tion beyond conventional biomarkers, we applied cPCA to a set of 21 CKD-related phenotypes. Initially, we generated 2,097,130 unique combinations out of the 21 phenotypes. These combinations encompassed all possible subsets of the 21 phenotypes with varying numbers, ranging from 2 to the complete set of 21. For each combination, we extracted the first principal component as CP, resulting in 2 million CPs.

Here, we only select the first principal component (PC1) because of several reasons. First, PC1 represents the direc-tion of maximum variance within the biomarker combination, capturing the most substantial and dominant patterns of variability in the data. In disease classification, capturing this primary variance is crucial, as it likely reflects the strongest underlying biological signals linked to disease risk. Higher-order principal components (e.g., PC2, PC3) often capture less significant variance, which may be more influenced by noise, measurement error, or irrelevant biological variation. By focusing on PC1, we emphasize the most reliable signal, which could improve the robustness of the disease classification model. Additionally, by reducing the dimensionality to PC1, the method simplifies the analysis, making it computationally more efficient, especially when the number of generated combinations is large.

Subsequently, we evaluated the performance of each CP in CKD classification and compared it to that of CYSC, which served as the best single marker for CKD classification. To validate the efficacy of the identified combinations, we parti-tioned the dataset into a training set (70%) and a test set (30%). Notably, cPCA was exclusively performed on the training set, encompassing the 2 million combinations. The performance evaluation involved comparing the ROC curves (Receiver Operating Characteristic curves) of each CP against those of individual phenotypes. Confidence intervals and derived p-values for the calculated AUCs (Area Under the Curve) were computed using bootstrap methods with 2000 stratified bootstrap replicates, implemented within the R package pROC [49]. Combinations exhibiting significantly higher AUCs compared to eGFR were further validated using the independent test set. The cPCA script written in R is available in S2 Text.

## Genome-wide analyses

Genome-wide association studies (GWAS) were performed by fitting linear models (for quantitative traits) or logistic models (for binary traits) implemented in PLINK2.0 [48]. All the input phenotypes were inverse-normal transformed prior to GWAS. Age, sex, and the first 20 genetic principal components were integrated into the models as covariates. SNP-based

heritability and genetic correlation were estimated based on the GWAS summary statistics using linkage disequilibrium score regression (LDSC) v1.0.1 [50].

## Colocalization analysis

To assess whether the loci identified through the multi-phenotype cPCA approach share causal variants with known eGFR-associated loci, we performed colocalization analysis using the coloc 5.2.3 R package [51]. This method estimates the probability that the same causal variant underlies both traits by evaluating their association signals at each locus.

For each locus identified in the cPCA GWAS and successfully replicated (*CST3* and *SH2B3*), we retrieved summary statistics from the eGFR GWAS and performed colocalization analysis across a ± 500 kb window centered on the lead SNP. The coloc Bayesian framework calculates the posterior probabilities for five hypotheses: no causal variant in either trait (PP0), causal variant in the composite phenotype GWAS only (PP1), causal variant in the eGFR GWAS only (PP2), two distinct causal variants (PP3), and one shared causal variant (PP4; evidence of colocalization). A locus was considered to exhibit strong colocalization if PP4 > 0.7, indicating a high probability that the same causal variant influences both traits.

## Supporting information

**S1 Table.** Trait loadings for the CP extracted from {eGFR, CYSC, ALB, HbA1c, GGT, LDL, and MA}.
(DOCX)

**S1 Text.** Method for handling urine albumin measurements below the detection limit.
(DOCX)

**S2 Text.** Scripts for conducting cPCA and downstream genomics analyses.
(DOCX)

## Acknowledgments

This research has been conducted using the UK Biobank Resource under Application Number 86460.

## Author contributions

**Conceptualization:** Kim Ngan Tran, Lyn R. Griffiths, Rodney Lea.

**Data curation:** Kim Ngan Tran.

**Formal analysis:** Kim Ngan Tran.

**Funding acquisition:** Lyn R. Griffiths.

**Investigation:** Kim Ngan Tran, Rodney Lea.

**Methodology:** Kim Ngan Tran, Andrew J. Mallett, Rodney Lea.

**Project administration:** Lyn R. Griffiths.

**Supervision:** Lyn R. Griffiths, Rodney Lea.

**Validation:** Heidi G. Sutherland, Andrew J. Mallett.

**Visualization:** Kim Ngan Tran.

**Writing – original draft:** Kim Ngan Tran.

**Writing – review & editing:** Kim Ngan Tran, Heidi G. Sutherland, Andrew J. Mallett, Lyn R. Griffiths, Rodney Lea.

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
