## [Decision Letter · Decision Letter 0]

8 Jan 2025

PGENETICS-D-24-01415

New composite phenotypes enhance Chronic Kidney Disease classification and genetic associations

PLOS Genetics

Dear Dr. Lea,

Thank you for submitting your manuscript to PLOS Genetics. After careful consideration, we feel that it has merit but does not fully meet PLOS Genetics's publication criteria as it currently stands. Therefore, we invite you to submit a revised version of the manuscript that addresses the points raised during the review process.

Please submit your revised manuscript within 60 days Mar 09 2025 11:59PM. If you will need more time than this to complete your revisions, please reply to this message or contact the journal office at plosgenetics@plos.org. Please include the following items when submitting your revised manuscript:

We look forward to receiving your revised manuscript.

Kind regards,

David A. Buchner, PhD

Academic Editor

PLOS Genetics

Zoltán Kutalik

Section Editor

PLOS Genetics

Aimée Dudley

Editor-in-Chief

PLOS Genetics

Anne Goriely

Editor-in-Chief

PLOS Genetics

**Journal Requirements:**

4) Please amend your detailed Financial Disclosure statement. This is published with the article. It must therefore be completed in full sentences and contain the exact wording you wish to be published.

**Reviewers' comments:**

Reviewer's Responses to Questions

**Comments to the Authors:**

Reviewer #1: I commend the authors on this undertaking and for writing a clear and concise article looking at combinations of biomarkers and genetic hits in CKD. It was a pleasure to read.

While the findings themselves may not be that unexpected, the approach of using PCA to look at biomarkers in a combinatorial manner is a 'why didn't I think of that' moment. It's straightforward, clear, and addresses a lot of issues that arise from using more complex machine learning approaches and should have broad applicability across other large cohort studies - and perhaps in smaller, deeply phenotyped ones as well.

Reviewer #2: PGENETICS-D-2401415

The study by Tran et al. aims to address the limitations of using single biomarkers to assess Chronic Kidney Disease (CKD) by applying combinatorial Principal Component Analysis (cPCA). Using UK Biobank data, the investigators analyzed 21 CKD-related phenotypes, generating over 2 million composite phenotypes (CPs). The top-ranked CP achieving an AUC of 0.878, outperforming eGFR alone (AUC: 0.83). Genetic association analysis identified all major eGFR-associated loci and the SH2B3 locus rs318450, a loss-of-function variant. The study highlights the effectiveness of the cPCA approach in understanding CKD's genetic basis and suggests potential applications for other complex diseases.

This is a novel and interesting approach to dealing with multiple phenotypes. The application to the example of chronic kidney disease is appropriate, although the “gold-standard” definition used here is not conventional nor is it fully adequate. However, the method can be useful in settings such as these.

- The abstract states that the SH2B3 locus has a combined beta of -0.046, with no units associated. Please allow the reader to understand how to interpret that effect estimate without having to read the whole manuscript.

- Could the investigators also show how a plot similar to Figure 2 but with CKD instead of eGFR.

- It would be helpful to show how this approach improves upon other PCA methods, if at all. Also, what are the loadings of the different biomarkers in each PC?

- The “gold standard” against which the biomarker combinations are being tested is CKD defined by ICD-code. The ICD-codes chosen for this definition have notoriously low sensitivity (around 25-30%) and varying specificity. Because ICD codes for CKD underestimate the true incidence and prevalence of CKD, with moderate cases of CKD particularly underreported, this study may be inadvertently selecting individuals with more severe disease, which may limit the generalizability of the findings and should be acknowledged as a limitation. This may also explain why some of the loci found to be associated with CKD but not in the original eGFR GWAS are linked to comorbidities (e.g. FTO – obesity, SH2B3 - diabetes).

- Microalbuminuria is only available in 156K participants in the UK Biobank. Were the remaining 200K imputed? Please elaborate on the imputation details.

- Typo on page 5, line 90 – “non-significant”. Beyond the typo issue – it would be more helpful to give the actual p-value rather than p>0.05.

- Typo on page 12, line 178 – should be “eGFR” instead of CYSC, no?

Reviewer #3: Tran et al. proposed a combinatorial Principal Component Analysis (cPCA) to analyze the combination of different etiologies that lead to loss of kidney function. The top-ranked principal components showed higher classification power for clinical chronic kidney disease. However, it is not clear what are the biological insights delivered in this study. Some major issues are listed in the following:

1. The phenotypes used are not independent of each other, which may introduce bias in the quantification of the combinational effects of chronic kidney disease.

2. Does the study identify a new variant associated with kidney function or chronic kidney disease? For example, the example variant (rs3184504, there is a typo rs318450 in the abstract) has been identified as a variant associated with eGFR in previous GWAS studies.

3. The conclusions in this study need to be validated by independent methods. For example, colocalization analysis of multiple traits is a powerful tool for exploring the combinational contribution of multiple traits. It would be useful to compare the preformation of cPCA and colocalization analysis with this aim.

4. What’s the rule of the order (No.) of phonotypes is in Table 1? It would be useful to order the traits based on the AUC.

**Have all data underlying the figures and results presented in the manuscript been provided?**

Reviewer #1: Yes

Reviewer #2: Yes

Reviewer #3: Yes

PLOS authors have the option to publish the peer review history of their article (what does this mean? ). If published, this will include your full peer review and any attached files.

**Do you want your identity to be public for this peer review?** For information about this choice, including consent withdrawal, please see our Privacy Policy .

Reviewer #1: No

Reviewer #2: No

Reviewer #3: No

**Figure resubmission:**
---

## [Decision Letter · Decision Letter 1]

10 Apr 2025

PGENETICS-D-24-01415R1

New composite phenotypes enhance chronic kidney disease classification and genetic associations

PLOS Genetics

Dear Dr. Lea,

Thank you for submitting your manuscript to PLOS Genetics. After careful consideration, we feel that it has merit but does not fully meet PLOS Genetics's publication criteria as it currently stands. Therefore, we invite you to submit a revised version of the manuscript that addresses the points raised during the review process.

Please submit your revised manuscript within 30 days May 10 2025 11:59PM. If you will need more time than this to complete your revisions, please reply to this message or contact the journal office at plosgenetics@plos.org. Please include the following items when submitting your revised manuscript:

We look forward to receiving your revised manuscript.

Kind regards,

David A. Buchner, PhD

Academic Editor

PLOS Genetics

Zoltán Kutalik, PhD

Section Editor

PLOS Genetics

Aimée Dudley

Editor-in-Chief

PLOS Genetics

Anne Goriely

Editor-in-Chief

PLOS Genetics

**Additional Editor Comments :**

In order to be able to accept the manuscript, we would like the remaining comment of reviewer #2 to be transparently addressed: for each identified variant check whether they have been picked up in previous CKD GWAS studies, even if bigger ones. We are convinced that the method is more powerful than standard GWAS of the same size, but still the readers deserve to know which of these identified SNPs (or their LD proxies) have been previously identified in larger studies. This would also be a nice validation of the findings.

**Reviewers' comments:**

Reviewer's Responses to Questions

Reviewer #2: I would like to commend the authors for the thorough and responsive resubmission. I have no further comments.

Reviewer #3: Thank you for answer the comments.

The only remained concern is "2. Does the study identify a new variant associated with kidney function or chronic kidney disease?". If there is new variant, please list them. If not, please also list the previous studies which have identified each variant, as an validation. This information is necessary to illustrate the importance of these significant variants.

**Have all data underlying the figures and results presented in the manuscript been provided?**

Reviewer #2: Yes

Reviewer #3: Yes

PLOS authors have the option to publish the peer review history of their article (what does this mean? ). If published, this will include your full peer review and any attached files.

**Do you want your identity to be public for this peer review?** For information about this choice, including consent withdrawal, please see our Privacy Policy .

Reviewer #2: **Yes: ** Cassianne Robinson-Cohen

Reviewer #3: No

**Figure resubmission:**
---

## [Editor Report · Decision Letter 2]

9 May 2025

Dear Dr Lea,

We are pleased to inform you that your manuscript entitled "New composite phenotypes enhance chronic kidney disease classification and genetic associations" has been editorially accepted for publication in PLOS Genetics. Congratulations!

Yours sincerely,

David A. Buchner, PhD

Academic Editor

PLOS Genetics

Zoltán Kutalik

Section Editor

PLOS Genetics

Aimée Dudley

Editor-in-Chief

PLOS Genetics

Anne Goriely

Editor-in-Chief

PLOS Genetics

Comments from the reviewers (if applicable):

**Data Deposition**

http://datadryad.org/submit?journalID=pgenetics&manu=PGENETICS-D-24-01415R2

**Press Queries**

---

## [Editor Report · Acceptance letter]

PGENETICS-D-24-01415R2

New composite phenotypes enhance chronic kidney disease classification and genetic associations

Dear Dr Lea,

We are pleased to inform you that your manuscript entitled "New composite phenotypes enhance chronic kidney disease classification and genetic associations" has been formally accepted for publication in PLOS Genetics! Your manuscript is now with our production department and you will be notified of the publication date in due course.

With kind regards,

Zsofia Freund

PLOS Genetics

On behalf of:
